# UnifiedSSR: A Unified Framework of Sequential Search and Recommendation

Submission Id: 647

## ABSTRACT

In this work, we propose a Unified framework of Sequential Search and Recommendation (UnifiedSSR) for joint learning of user behavior history in both search and recommendation scenarios. Specifically, we consider user-interacted products in the recommendation scenario, user-interacted products and user-issued queries in the search scenario as three distinct types of user behaviors. We propose a dual-branch network to encode the pair of interacted product history and issued query history in the search scenario in parallel. This allows for cross-scenario modeling by deactivating the query branch for the recommendation scenario. Through the parameter sharing between dual branches, as well as between product branches in two scenarios, we incorporate cross-view and cross-scenario associations of user behaviors, providing a comprehensive understanding of user behavior patterns. To further enhance user behavior modeling by capturing the underlying dynamic intent, an Intent-oriented Session Modeling module is designed for inferring intent-oriented semantic sessions from the contextual information in behavior sequences. In particular, we consider self-supervised learning signals from two perspectives for intent-oriented semantic session locating, which encourages session discrimination within each behavior sequence and session alignment between dual behavior sequences. Extensive experiments on three public datasets demonstrate that UnifiedSSR consistently outperforms state-of-the-art methods for both search and recommendation.

## CCS CONCEPTS

• **Information systems** → **Personalization**; **Learning to rank**; **Recommender systems**.

## KEYWORDS

Personalized Search and Recommendation; Sequential User Behavior Modeling; Multi-Task Learning; Joint Learning; E-Commerce

ACM Reference Format:
Anonymous Author(s). 2018. UnifiedSSR: A Unified Framework of Sequential Search and Recommendation. In *Proceedings of Make sure to enter the correct conference title from your rights confirmation email (Conference acronym 'XX)*. ACM, New York, NY, USA, 10 pages. https://doi.org/XXXXXXX.XXXXXXX

## 1 INTRODUCTION

On e-commerce platforms, users typically interact with products in two major scenarios, *i.e.*, search and recommendation. Users can either interact directly with products listed on the recommendation page, or issue a query in the search box and then proceed to interact with products displayed on the search result page. For a long time, search and recommendation have been regarded as two separate research scenarios, each becoming increasingly prevalent in real-world applications. Recommendation engines mine user preferences from behavior history to suggest personalized products [6, 29], while search engines assist users in finding specific products based on their queries [18, 21]. A key distinction between the search and recommendation scenarios lies in the fact that users provide explicit queries for search, whereas no query is present for recommendation. Nevertheless, in both scenarios, the goal of the models is to generate a personalized ranked list of products, which satisfies the personalized needs of users and alleviates information overload. Despite the recent success achieved by studies in each individual scenario, they still face challenges related to limited representation capabilities and data sparsity issues [29, 35].

Figure 1 depicts an overview of the connections between the search and recommendation in an integrated system, where the user set, product set, and vocabulary are shared. Despite the use of different techniques in search and recommendation engines, the two scenarios are closely related, and therefore, learning in one scenario may potentially benefit the other. In this sense, leveraging user behavior data from both scenarios to construct a unified model holds the potential for mutual enhancement in user modeling. The joint learning of a unified model helps alleviate data sparsity issues while simultaneously improving model performance in both scenarios, eventually contributing to the overall user satisfaction.

Pioneering studies [31, 33–35] have demonstrated the superiority of unified models over single-scenario models in both search and recommendation. However, these methods either simply combine individual models for the two tasks through a joint loss function [33, 34], ignoring the correlation of user behaviors in both scenarios, or they treat user behaviors in the recommendation scenario as special cases in the search scenario with empty queries [31, 35], overlooking the inherent differences between user behaviors in the two scenarios. Different from these approaches, in this work, we aim to construct a unified model that effectively leverages the commonalities and differences across user behaviors in both search and recommendation. To achieve this, the following two challenges should be considered:

**Challenge 1: Cross-scenario and cross-view user behavior modeling**. Users engage in three distinct behavior types across scenarios: (a) *interacting with products* in the recommendation scenario, (b) *issuing queries* and then (c) *interacting with products* in the search scenario. In the recommendation scenario, users interact with products without a clear intent, whereas they interact with

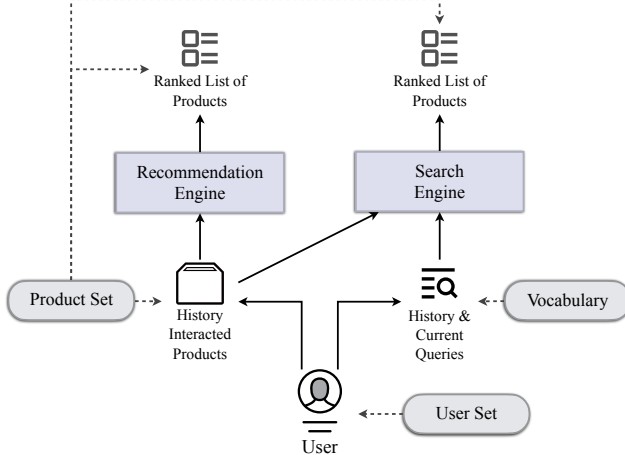

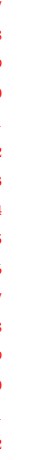

**Figure 1: An overview of the system architecture for the integrated personalized search and recommendation within an e-commerce platform, where the user set, product set, and vocabulary are shared. Note that side information such as user profiles and product metadata has been excluded for simplicity.**

products driven by a specific intent in the search scenario. Consequently, the product interaction histories in these two scenarios may exhibit different distributions. Thus, it is important to account for the commonalities and differences in cross-scenario product interactions to construct a unified model. Furthermore, in the search scenario, users explicitly express their intent through natural language queries, and then selectively interact with products from search results. The pair of issued query and interacted product can be regarded as two views on user intent. The issued query provides more informative insights into user intent but is more difficult to learn due to its unstructured nature. On the contrary, the interacted product is easier to model but may not always be reliable due to exposure bias [27]. Hence, it is also crucial to consider the commonalities and differences in cross-view user behaviors.

**Challenge 2: Joint dynamic user intent modeling**. Another significant challenge lies in uncovering the underlying user intent behind each interaction in a long history sequence. Since user intent evolves over time, users engage in a series of consecutive behaviors driven by specific or broad intent, after which that intent may drift or even abruptly change for various reasons [5]. Discovering and aggregating semantic sessions resulting from distinct intents is beneficial for enhancing user intent understanding in user behavior modeling. However, how to effectively locate the intent-oriented sessions with variable lengths remains unexplored.

To address the aforementioned challenges, we propose a Unified framework of Sequential Search and Recommendation (UnifiedSSR) for joint learning of user behavior history in both search and recommendation scenarios. First, we propose a dual-branch network to encode the pair of interacted product history and issued query history in the search scenario, and deactivate the query branch to adapt to the recommendation scenario. Through the parameter sharing between dual branches, as well as between product

branches in two scenarios, our unified model effectively shares information cross-scenario (*i.e.*, search and recommendation scenarios) and cross-view (*i.e.*, interacted products and issued queries in the search scenario), resulting in a comprehensive understanding of user behavior patterns. Second, in order to enhance user behavior modeling by leveraging dynamic user intent, an Intent-oriented Session Modeling module is designed that discovers intent-oriented semantic sessions based on the contextual information in behavior sequences. In particular, we utilize two self-supervised learning signals based on similarity measurements for intent-oriented semantic session discovery: (1) Sessions resulting from different user intents within each behavior sequence should be distinguished from each other. Therefore, we facilitate the distinction between adjacent intent-oriented sessions in each behavior sequence. (2) When a user interacts with a product after issuing a query, this pair of interacted product and issued query driven by a common intent should align with each other. Consequently, we promote the alignment of the pair of interacted product session and issued query session guided by the same intent in dual behavior sequences.

Our contributions in this work can be summarized as follows:

- We propose a new Unified framework for Sequential Search and Recommendation (UnifiedSSR), which employs a dual-branch architecture with shared parameters to enable the joint learning of cross-scenario cross-view user behaviors.
- We design an Intent-oriented Session Modeling module to enhance user behavior modeling by capturing the dynamic user intent. Particularly, two self-supervised learning signals are leveraged that encourage intent-oriented session discrimination within each behavior sequence and intent-oriented session alignment between dual behavior sequences.
- We conduct extensive experiments on three public datasets. The experimental results demonstrate that UnifiedSSR outperforms state-of-the-art joint models and scenario-specific models in both search and recommendation scenarios.

## 2 RELATED WORK

Recent years have witnessed significant success of research in each individual domain of search [3, 8, 18, 19, 21] and recommendation [6, 20, 29, 30], leading to a substantial amount of outstanding work. However, to the best of our knowledge, rarely have efforts been dedicated to joint modeling of search and recommendation. We broadly classify these pioneering studies into two categories: search data enhanced recommender systems [13, 23, 24, 28] and multi-scenario unified models [31, 33–35]. Search data enhanced recommender systems treat user behaviors in the search scenario as complementary information to boost the recommendation performance. For instance, NRHUB [28] utilized a hierarchical attention-based multi-view encoder to learn unified representations of users from their heterogeneous behaviors, including search query behaviors. Query-SeqRec [13] directly constructed query-aware heterogeneous sequences that contain both query interactions and item interactions, based on which the next interacted item is predicted. IV4Rec [23] leveraged search queries as instrumental variables to decompose and reconstruct user and item embeddings in a causal learning manner. SESRec [24] disentangled similar and dissimilar representations

 

in both search and recommendation behaviors, achieving comprehensive recommendation based on multiple aspects. These models exploit search data to enhance recommendation performance, neglecting the potential for combining two scenarios to complement each other and jointly improve the model performance in both scenarios.

On the other hand, multi-scenario unified models perform joint learning of search and recommendation to enhance the model performance in both scenarios. JSR [33] simultaneously trained two MLP-based models for search and recommendation using a joint loss function. Experimental results demonstrate that the joint model substantially outperforms the independently trained models for each scenario. Afterwards, JSR was extended by incorporating relevance-based word embedding [32] into the search model and neural collaborative filtering [12] into the recommendation model [34]. These models merely combined two scenario-specific models through a joint loss function, failing to account for the intrinsic correlations between user behaviors in two scenarios. More recently, USER [31] adopted a hierarchical structure, using Transformer in three levels to encode heterogeneous sequences consisting of queries and interacted documents. SRJGraph [35] constructed a unified graph from both search and recommendation data, where users and items are heterogeneous nodes and search queries are incorporated into the user-item interaction edges as attributes. Both USER and SRJGraph fuse interactions in two scenarios by regarding user behaviors in the recommendation scenario as special cases in the search scenario with an empty query. These models effectively model the commonalities between the two scenarios but overlooked their distinct characteristics. Instead, we propose a unified framework that effectively leverages the commonalities and differences in cross-view cross-scenario user behaviors.

## 3 METHODOLOGY

### 3.1 Problem Statement

Let $\mathcal{U}$, $\mathcal{P}$, $\mathcal{Q}$ denotes the sets of users, products and queries, respectively. For each user $u \in \mathcal{U}$, interactions with products $p \in \mathcal{P}$ occur in both search and recommendation scenarios, with each interaction conveying the user intent and preference. In both scenarios, the product sequence in chronological order of user $u$ can be denoted as $\mathcal{S}^p = \{p_t \mid t = 1, 2, \ldots, T\}$, where $p_t \in \mathcal{P}$ is the interacted product at timestep $t$. We use $\mathcal{S}_s^p = \{p_{t_s}\}$ and $\mathcal{S}_r^p = \{p_{t_r}\}$ to distinguish product sequences in search and recommendation scenarios, respectively. In the search scenario, we incorporate the issued queries through an additional query sequence $\mathcal{S}_s^q = \{q_{t_s} \mid t_s = 1, 2, \ldots, T_s + 1\}$ for user $u$, where the query $q_{t_s} \in \mathcal{Q}$ is composed of a series of words $\{w_1, w_2, \ldots, w_{|q_{t_s}|}\}$ from the word vocabulary $\mathcal{V}$. The product sequence and query sequence are synchronized in timestep, namely, the pair $\langle p_{t_s}, q_{t_s} \rangle$ represents that user $u$ interacts with product $p_{t_s}$ from the search result page of issued query $q_{t_s}$ at timestep $t$. Besides, $q_{T_s+1}$ is the issued query for the product search in the next timestep.

Given a user $u$ with historical behavior sequences, the unified model aims to predict whether the user will interact with a product $p$ when it is exposed to them in the next timestep, in either the search or recommendation scenario. Specifically, the model objectives in both scenarios can be holistically formulated as estimating

personalized ranking scores for products by:

$$\hat{y}_{u,p} = \begin{cases} f_\Theta(p_{T_r+1} \mid u, S_r^p), & \text{if recommendation,} \\ f_\Theta(p_{T_s+1} \mid u, S_s^p, S_s^q), & \text{otherwise.} \end{cases} \quad (1)$$

$f_\Theta(\cdot)$ denotes the underlying unified model with parameters $\Theta$, and $\hat{y}_{u,p}$ is the predicted score for product $p$ that user $u$ is likely to interact with in the next timestep. The top-$K$ products ranked by predicted scores are the final results provided by the model.

### 3.2 Overall Architecture

The UnifiedSSR framework is illustrated in Figure 2. It consists of two branches, i.e., product branch and query branch. Two branches share parameters to transform two types of sequences into a common latent space, allowing UnifiedSSR to simultaneously learn user behavior patterns across two views. Due to the overall dual-branch architecture, the product sequence learning in the recommendation scenario can be directly achieved by deactivating the query branch, thereby enabling cross-scenario joint learning of the model. Overall, the information sharing characteristics of UnifiedSSR are manifested in two aspects: (1) the shared parameters for representation learning of product sequences in both scenarios; (2) the shared parameters for representation learning of the product sequence and query sequence in the search scenario.

Taking the search data as an example, the **Embedding Module** embeds the pair of product sequence and query sequence into dense representations, followed by a parameter-shared **Siamese Encoder** that comprehensively captures the correlations both within and between dual behavior sequences. Next, an **Intent-oriented Session Modeling** is proposed to locate intent-oriented semantic sessions, obtaining representations of these sessions to enhance sequence representation matrices. In particular, a self-supervised learning loss function based on similarity measurements is designed, which guides the intent-oriented session discovery by encouraging session discrimination within each sequence and session alignment across dual sequences. The intent-enhanced sequence representations are then fed into the final **Task-specific Predictor** to obtain the predicted results for different scenarios. The details of UnifiedSSR are described as follows.

### 3.3 Embedding Module

In the embedding module, high-dimensional one-hot representations of users, products and query words are transformed into dense representations of dimension $d$ through embedding matrices $\mathbf{M}^u \in \mathbb{R}^{|\mathcal{U}| \times d}$, $\mathbf{M}^p \in \mathbb{R}^{|\mathcal{P}| \times d}$, $\mathbf{M}^w \in \mathbb{R}^{|\mathcal{V}| \times d}$. While a query comprises a series of words, it is typically short and lacks sequential patterns [5]. Therefore, the embedding of a query $q = \{w_1, w_2, \ldots, w_{|q|}\}$ can be effectively obtained by performing mean pooling on word embeddings as: $\mathbf{e}^q = \text{Mean}(\mathbf{e}^{w_1}, \mathbf{e}^{w_2}, \ldots, \mathbf{e}^{w_{|q|}})$, where $\mathbf{e}^{w_i}$ is the embedding of $i$-th word in the query.

Given a product sequence $\mathcal{S}^p$ with a length of $T$ in either the search or recommendation scenario, we obtain its sequence embedding matrix as $\mathbf{E}^p = [\mathbf{e}_1^p + \mathbf{e}^u; \mathbf{e}_2^p + \mathbf{e}^u; \ldots; \mathbf{e}_T^p + \mathbf{e}^u] \in \mathbb{R}^{T \times d}$, where $\mathbf{e}_t^p$ denotes the embedding of product $p$ at timestep $t$, and $\mathbf{e}^u$ denotes the embedding of user $u$. Besides, we add positional encodings $\mathbf{P}$ to $\mathbf{E}^p$, i.e., $\mathbf{E}^p = \mathbf{E}^p + \mathbf{P}$ to inject the relative positional information into the sequence embedding matrix [26]. For the query sequence $\mathcal{S}_s^q$ of

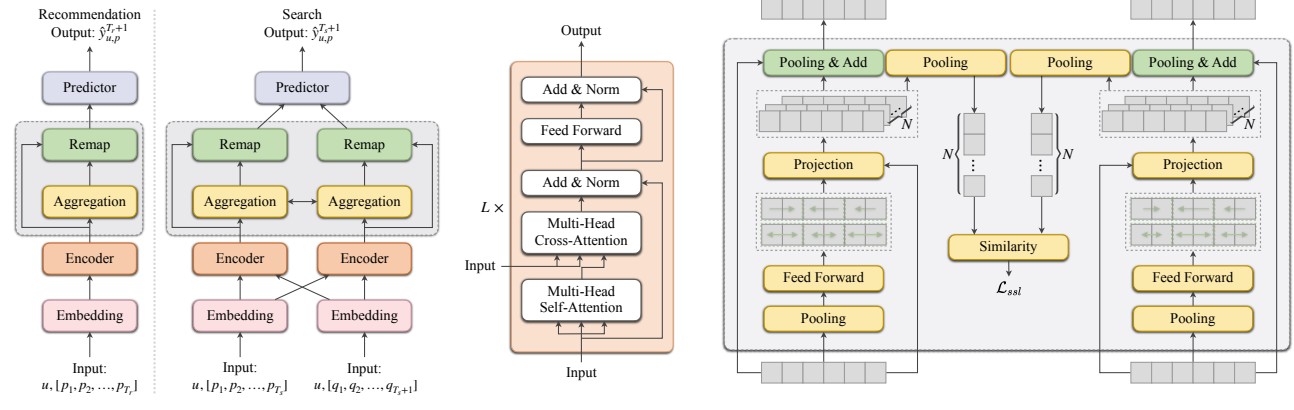

(a) UnifiedSSR      (b) Siamese Encoder Layer      (c) Intent-oriented Session Modeling

**Figure 2: An overview of the proposed UnifiedSSR framework. (a) presents the architecture of UnifiedSSR with query branch deactivated for recommendation (left) and with entire dual branches for search (right). The information sharing mechanism is two-fold: cross-scenario parameter sharing for learning user-interacted products in two scenarios, cross-view parameter sharing for learning user-interacted products and user-issued queries in the search scenario. (b) illustrates the structure of the Siamese Encoder layer. (c) demonstrates the complete Intent-oriented Session Modeling in the search scenario.**

length $(T+1)$, we compute the sequence embedding matrix $\mathbf{E}_s^q$ in a similar manner, *i.e.*, $\mathbf{E}_s^q = [\mathbf{e}_1^q + \mathbf{e}^u; \mathbf{e}_2^q + \mathbf{e}^u; \ldots; \mathbf{e}_{T+1}^q + \mathbf{e}^u] + \mathbf{P} \in \mathbb{R}^{(T+1) \times d}$. For clarity, we denote the embedding matrices of the product sequence in the recommendation scenario, the product and query sequences in the search scenario as $\mathbf{E}_r^p$, $\mathbf{E}_s^p$, $\mathbf{E}_s^q$, respectively.

### 3.4 Siamese Encoder

In the search scenario, the user-issued query and user-interacted product at each timestep are different types of behaviors driven by a common user intent. In order to encode these two behavior sequences while leveraging their common and unique characteristics, we propose a Siamese Encoder with shared parameters that takes two sequences as pairs to be encoded in parallel. The Siamese Encoder encodes correlations both within and between the product sequence and query sequence in the search scenario, while encodes correlations within the product sequence in the recommendation scenario. As such, the Siamese Encoder is capable of learning a comprehensive representation of sequential user behavior patterns.

Inspired by the encoder layer in the vanilla Transformer [26], the Siamese Encoder layer is designed to contain three sub-layers, *i.e.*, the Multi-head Self-Attention (MSA), Multi-head Cross-Attention (MCA), and Feed-Forward Network (FFN).

We briefly review the Multi-head Attention (MA) mechanism with the scaled dot-product attention, which can be described as follows:

$$\text{MA}(\mathbf{Q}, \mathbf{K}, \mathbf{V}) = \text{Concat}([\text{Attn}_1; \text{Attn}_2; \cdots; \text{Attn}_h])\mathbf{W}^O,$$

$$\text{Attn}_i(\mathbf{Q}\mathbf{W}_i^Q, \mathbf{K}\mathbf{W}_i^K, \mathbf{V}\mathbf{W}_i^V) = \text{softmax}(\frac{(\mathbf{Q}\mathbf{W}_i^Q)(\mathbf{K}\mathbf{W}_i^K)^T}{\sqrt{d_h}})(\mathbf{V}\mathbf{W}_i^V).$$
(2)

The projection matrices $\mathbf{W}_i^Q \in \mathbb{R}^{d \times d_h}$, $\mathbf{W}_i^K \in \mathbb{R}^{d \times d_h}$, $\mathbf{W}_i^V \in \mathbb{R}^{d \times d_h}$, $\mathbf{W}^O \in \mathbb{R}^{d \times d}$ are learnable parameters, where $h$ is the number of attention heads, and $d_h = d/h$.

In the case of the product branch in the search scenario, the multi-head self-attention operation focuses on the correlation within the sequence, which takes the embedding matrix $\mathbf{E}_s^p$ as the input of MA, *i.e.*, $\mathbf{Q} = \mathbf{K} = \mathbf{V} = \mathbf{E}_s^p$. Then, the multi-head cross-attention is followed to encode the correlation across two sequences. Specifically, the multi-head cross-attention take both $\mathbf{E}_s^p$ and $\mathbf{E}_s^q$ as input of MA, *i.e.*, $\mathbf{Q} = \mathbf{E}_s^p$, $\mathbf{K} = \mathbf{V} = \mathbf{E}_s^q$.

After encoding the intra- and inter-correlations of sequences, a position-wise feed-forward network is then applied, consisting of two linear transformations with a ReLU activation in between.

The Siamese Encoder layer comprehensively encodes the contextual information in dual behavior sequences, producing contextual representation matrices $\mathbf{H}_s^p$ and $\mathbf{H}_s^q$ for the product and query sequences, respectively. This can be summarized as follows:

$$\hat{\mathbf{H}}_s^p = \text{MSA}(\mathbf{E}_s^p, \mathbf{E}_s^p, \mathbf{E}_s^p), \quad \hat{\mathbf{H}}_s^q = \text{MSA}(\mathbf{E}_s^q, \mathbf{E}_s^q, \mathbf{E}_s^q),$$
$$\mathbf{H}_s^p = \text{FFN}(\text{MCA}(\hat{\mathbf{H}}_s^p, \hat{\mathbf{H}}_s^q, \hat{\mathbf{H}}_s^q)), \quad (3)$$
$$\mathbf{H}_s^q = \text{FFN}(\text{MCA}(\hat{\mathbf{H}}_s^q, \hat{\mathbf{H}}_s^p, \hat{\mathbf{H}}_s^p)),$$

where $\text{MSA}(\cdot)$, $\text{MCA}(\cdot)$, $\text{FFN}(\cdot)$ denote the aforementioned three sub-layers. Note that we also adopt the residual connection [11], layer normalization [4], and dropout regularization [25] to enhance the network structure following [15, 26].

For the recommendation scenario where user-issued queries are absent, the Siamese Encoder layer can be adapted by deactivating the query branch. As such, the multi-head cross-attention becomes equivalent to the multi-head self-attention, and the contextual representation matrix of the product sequence is derived as:

$$\hat{\mathbf{H}}_r^p = \text{MSA}(\mathbf{E}_r^p, \mathbf{E}_r^p, \mathbf{E}_r^p), \quad \mathbf{H}_r^p = \text{FFN}(\text{MCA}(\hat{\mathbf{H}}_r^p, \hat{\mathbf{H}}_r^p, \hat{\mathbf{H}}_r^p)). \quad (4)$$

After being encoded by the Siamese Encoder composed of a stack of $L$ identical layers, we obtain the contextual representation matrices for the product and query sequences in the search scenario, denoted as $\mathbf{H}_s^p$ and $\mathbf{H}_s^q$, and for the product sequence in the recommendation scenario, represented as $\mathbf{H}_r^p$. Here the superscript

($L$) indicating the number of Siamese Encoder layers is omitted for simplicity.

## 3.5 Intent-oriented Session Modeling

Leveraging the inherent user intent associated with each interaction could potentially improve the user behavior modeling. In most cases, however, there is no labeled data explicitly revealing the intent for each interaction. Since user intent evolves over time, users engage in a series of consecutive behaviors driven by one intent, followed by another series of consecutive behaviors under a different intent. Accordingly, we propose an Intent-oriented Session Modeling module, which captures user intent by locating and aggregating intent-oriented semantic sessions based on the contextual information in behavior sequences, so as to achieve intent-enhanced user behavior modeling. In particular, a self-supervised learning loss based on similarity measurements is designed to guide the intent-oriented session discovery. In this section, we mainly use the search scenario as an example to introduce the Intent-oriented Session Modeling module, so we omit the subscript $s/r$ distinguishing search and recommendation scenarios to simplify the notation.

*3.5.1 Intent-oriented Session Extraction.* In the case of the product sequence, it is first uniformly divided into $N$ non-overlapping sessions. Let $\mathbf{x} = [x_1, x_2, \ldots, x_N]$ represent central locations of sessions in the sequence $\mathcal{S}_p$, where $x_i$ denotes the central location of the $i$-th session. The session location ranges are initialized as $(\mathbf{x} - \frac{L}{2N}, \mathbf{x} + \frac{L}{2N})$, thereby the sequence representation matrix can be sliced into chunks as $\mathbf{H}^p = [\mathbf{H}^p_{\mathbf{s}_1}; \mathbf{H}^p_{\mathbf{s}_2}; \ldots; \mathbf{H}^p_{\mathbf{s}_N}]$. In order to locate intent-oriented sessions, we make the session location ranges learnable, which can be inferred from the contextual representation matrix of the behavior sequence. In particular, inspired by [7] for semantic patch learning in vision tasks, we predict offsets $\Delta\mathbf{x}$ of central locations and lengths $\mathbf{s}$ based on the contextual representation matrix $\mathbf{H}^p$ as follows:

$$\begin{aligned} \Delta\mathbf{x} &= \mathrm{Tanh}(f(\mathbf{H}^p)), \\ \mathbf{s} &= \mathrm{ReLU}(\mathrm{Tanh}(f(\mathbf{H}^p) + \mathbf{b})), \end{aligned} \quad (5)$$

where $f(\cdot)$ denotes the transformation that deduces the offset and length from the sequence representation matrix. We implement the transformation as a concatenation of mean pooling for each chunked representation matrix, followed by a linear transformation with a ReLU activation in between, which can be written as:

$$f(\mathbf{H}^p) = \mathrm{ReLU}(\mathrm{Concat}[\mathrm{Mean}(\mathbf{H}^p_{\mathbf{s}_1}); \ldots; \mathrm{Mean}(\mathbf{H}^p_{\mathbf{s}_N})])\mathbf{W}. \quad (6)$$

Accordingly, the $i$-th intent-oriented session is updated to be located in $(x_i + \Delta x_i - s_i, x_i + \Delta x_i + s_i)$. In this way, we can fully exploit the context to identify semantic sessions. We use $(\mathbf{x}^{\mathrm{left}}, \mathbf{x}^{\mathrm{right}})$ to denote the overall learned session ranges. After locating $N$ sessions in the product sequence, we then aggregate the interaction representations within each session, represented as $\{\mathcal{I}^p_i \mid 1 \leq i \leq N\}$, where $\mathcal{I}^p_i = \{\mathbf{H}^p_j \mid x^{\mathrm{left}}_i \leq j < x^{\mathrm{right}}_i\}$. As such, the session representation matrix $\mathbf{I}^p \in \mathbb{R}^{N \times d}$ can be derived by applying mean pooling to its containing interaction representations as follows:

$$\mathbf{I}^p = \mathrm{Concat}([\mathrm{Mean}(\mathcal{I}^p_1); \mathrm{Mean}(\mathcal{I}^p_2); \ldots; \mathrm{Mean}(\mathcal{I}^p_N)]), \quad (7)$$

where the $i$-th row in $\mathbf{I}^p$ represents the $i$-th intent-oriented session representation.

The representation of each interaction $\mathbf{H}^p_t$ is enhanced by integrating intent-oriented session representations as follows:

$$\mathbf{F}^p_t = \mathbf{H}^p_t + \sum_{i=1}^{N} \mathbf{I}^p_i \cdot \mathbb{I}[\mathbf{H}^p_t \in \mathcal{I}^p_i], \quad (8)$$

where $\mathbb{I}[\cdot]$ is an indicator function that returns 1 when the condition holds, and 0 otherwise.

Analogously, the representation matrix of the query sequence in the search scenario is also enhanced by aggregating intent-oriented session representations. Ultimately, we obtain the intent-enhanced contextual representation matrices $\mathbf{F}^p_r$ for the product sequence in the recommendation scenario, $\mathbf{F}^p_s$ and $\mathbf{F}^q_s$ for product and query sequences, respectively.

*3.5.2 Self-supervised Intent-oriented Session Discovery.* To further guide the intent-oriented session discovery, we consider two aspects of self-supervised signals: (1) Different user intents within a behavior sequence should lead to distinguishable sessions. Therefore, we encourage the representations of adjacent intent-oriented sessions within a sequence to be dissimilar to maintain discrimination. (2) A pair of product session and query session in dual behavior sequences driven by a common user intent should align with each other. Hence, we encourage the representations of corresponding intent-oriented sessions between two sequences to be similar to achieve alignment.

Accordingly, given session representation matrices $\mathbf{I}^p$ and $\mathbf{I}^q$ of product and query sequences in the search scenario, the self-supervised learning loss is defined as:

$$\mathcal{L}_{ssl} = \sum_{i=1}^{N-1} \left( \mathrm{Sim}(\mathbf{I}^p_i, \mathbf{I}^p_{i+1}) + \mathrm{Sim}(\mathbf{I}^q_i, \mathbf{I}^q_{i+1}) \right) - \sum_{i=1}^{N} \mathrm{Sim}(\mathbf{I}^p_i, \mathbf{I}^q_i), \quad (9)$$

where $\mathrm{Sim}(\cdot, \cdot)$ is the cosine similarity function. In Equation (9), the first term aims to minimize the similarity between adjacent semantic sessions to encourage the session discrimination within each of the two sequences, while the second term is designed to maximize the similarity between corresponding semantic sessions in two sequences to encourage the session alignment between two sequences.

As for the recommendation scenario with solely product interactions, the self-supervised learning loss simplifies to $\mathcal{L}_{ssl} = \sum_{i=1}^{N-1} \mathrm{Sim}(\mathbf{I}^p_i, \mathbf{I}^p_{i+1})$, guided by the first signal.

## 3.6 Task-specific Predictor

After the contextual information encoding and intent-oriented session enhancement, we obtain the behavior representations of each user as $\mathbf{f}^p_r \in \mathbb{R}^d$, $\mathbf{f}^p_s \in \mathbb{R}^d$, $\mathbf{f}^q_s \in \mathbb{R}^d$, corresponding to the last timestep of the representation matrices $\mathbf{F}^p_r$, $\mathbf{F}^p_s$, $\mathbf{F}^q_s$ of the product sequence in the recommendation scenario, product and query sequences in the search scenario, respectively. For the final prediction, two task-specific predictors are employed for search and recommendation tasks, respectively.

In the recommendation scenario, we adopt the widely used inner product [6, 36] to calculate the predicted score of the next interacted product $p$ as follows:

$$\hat{y}_{u,p} = \mathbf{f}^p_r \cdot \mathbf{e}^p, \quad (10)$$

where $\mathbf{e}^p$ is the embedding of product $p$ from the product embedding matrix $\mathbf{M}^p$.

Similarly, in the search scenario, we separately calculate the inner products for a given product $p$ with each of the two behavior representations, which are weighted and summed to derive the overall predicted score as follows:

$$\hat{y}_{u,p} = \left(w\mathbf{f}_s^p + (1-w)\mathbf{f}_s^q\right) \cdot \mathbf{e}^p, \tag{11}$$

where the balancing weight $w$ is a learnable parameter.

## 3.7 Model Optimization

We adopt the binary cross-entropy loss [15] to supervise the final prediction for both tasks as follows:

$$\mathcal{L}_{predict} = -\left[\log \sigma(\hat{y}_{u,p}) + \sum_{p^- \in \mathcal{P}_{neg}} \log(1 - \sigma(\hat{y}_{u,p^-}))\right], \tag{12}$$

where $\sigma(\cdot)$ is the sigmoid function, $\mathcal{P}_{neg}$ denotes the set of randomly sampled negative products paired with each ground-truth $p$.

The prediction and intent-oriented session discovery objectives are jointly optimized, forming the overall loss function as follows:

$$\mathcal{L}_{joint} = \mathcal{L}_{predict} + \alpha \cdot \mathcal{L}_{ssl}, \tag{13}$$

where $\alpha$ is a hyper-parameter that controls the weight of self-supervised learning loss for intent-oriented session discovery.

One of the core ideas behind UnifiedSSR is the integration of cross-scenario data to train a unified model, capitalizing on the commonalities and dependencies between search and recommendation scenarios. However, it is essential for the unified model not only to capture general patterns across scenarios but also to be tailored to specific tasks, ultimately leading to improved performance and robustness in both tasks. Accordingly, we adopt a training paradigm that consists of two stages: (1) *multi-task joint pre-training* and (2) *task-specific fine-tuning*. In particular, the entire framework is initially pretrained by alternately using data from two scenarios. Subsequently, for each task, the pretrained model is then finetuned individually using a small amount of task-specific data. As such, the model not only benefits from comprehensively training on cross-scenario data but also can be easily adapted to specific tasks.

## 4 EXPERIMENTS

### 4.1 Experimental Settings

*4.1.1 Datasets.* To evaluate the performance of UnifiedSSR in both search and recommendation scenarios, we conduct experiments on three publicly available datasets: JDsearch dataset [17], two subsets of Amazon review dataset [22], which are Clothing Shoes and Jewelry subset (referred to as Amazon-CL) and Electronics subset (referred to as Amazon-EL).

**JDsearch Dataset**: This dataset is a personalized product search dataset consisting of real user queries and user-product interactions collected from *JD.com*, one of the most popular Chinese e-commerce platforms. The dataset contains products belonging to various categories, interactions from diverse channels including search and recommendation, and all data have been anonymized. We extract the product interactions without corresponding queries from user

**Table 1: Statistics of Datasets**

|  | JDsearch | Amazon-CL | Amazon-EL |
|---|---|---|---|
| #Users | 131,701 | 323,714 | 192,586 |
| #Products | 411,566 | 393,214 | 180,446 |
| #QueryWords | 139,610 | 209,057 | 224,652 |
| #Interactions | 16,101,041 | 5,385,648 | 756,077 |
| #Samples-S | 126,179 | 162,023 | 96,529 |
| #Samples-R | 174,348 | 162,023 | 96,529 |

behavior logs to serve as recommendation data, with the remaining records treated as search data.

**Amazon Review Dataset**: This is a well-known dataset in recommender systems [15, 36], containing product reviews and metadata from *Amazon.com*. It is also the most commonly used public dataset in product search, featuring simulated queries derived from product metadata [2, 18]. We equally split the interaction history of each user into recommendation data and search data. Inspired by Gysel *et al.* [10], we use product categories, titles and brands to generate queries. Additionally, to introduce personalization into simulated queries, we extract keywords from user reviews based on TF-IDF, which are combined with product attributes to form the ultimate queries.

For each dataset, we filter out users and products with fewer than 10 interactions. The maximum sequence length of search and recommendation history is set to 100. Longer sequences are divided into non-overlapping subsequences. For both search and recommendation data, the sequences of each user are chronologically ordered and divided into subsets for multi-task joint learning and task-specific learning in an 8:2 ratio. The multi-task joint learning set is used for model pre-training, while the task-specific learning set is further split into training, validation, and test sets. In particular, the most recent interaction is reserved for testing, the second most recent interaction for validation, and all remaining interactions for training. The statistics of three datasets are summarized in Table 1.

*4.1.2 Baselines.* We compare the proposed UnifiedSSR with search models, recommendation models and joint models, as follows:

**Search Models**: (1) **HEM** [2] jointly learns different level embeddings of users, queries, products by maximizing the likelihood of observed user-query-product triplets to perform personalized product search. (2) **ZAM** [1] constructs query-dependent user embeddings based on an attention mechanism, introducing a zero vector in the attention operator to achieve differentiated personalization. (3) **CAMI** [18] builds upon the knowledge graph embedding method [3], leveraging the category information to disentangle and aggregate diverse interest embeddings of users.

**Recommendation Models**: (1) **GRU4Rec** [14] applies recurrent neural networks to model user interacted item sequences for session-based recommendation. (2) **SASRec** [15] directly implements the Transformer [26] encoder stacks with single-head self-attention mechanism for sequential recommendation. (3) **FMLP-Rec** [36] adopts all-MLP architecture derived from Transformer, where the attention mechanism is replaced with frequency-domain learnable filters.

**Joint Models**: (1) **JSR** [33] simultaneously learns two MLP-based models for retrieval and recommendation, based on a shared item

set and a joint loss function. (2) **JSR-Seq** is our extension of JSR, where the simple MLPs are replaced with our proposed sequential encoders. Note that the encoders share the same architecture but have separate parameters. (3) **SESRec** [24] employs Transformers to individually encode search and recommendation behaviors of users, disentangling similar and dissimilar representations between two behaviors to enhance recommendations. We integrate query embeddings into the prediction layer to adapt it to the search task.

Considering the two-stage training strategy adopted for our proposed UnifiedSSR, for a fair comparison, the above baselines utilized all available data for training, including both aforementioned pre-training and fine-tuning data. We also evaluate the performance of the proposed model end-to-end trained with task-specific data, represented as **UnifiedSSR-R** and **UnifiedSSR-S**, respectively. Besides, all methods share the same validation and test sets.

*4.1.3 Evaluation Metrics.* To evaluate the performance on both search and recommendation, we adopt two widely used evaluation protocols, Hit Ratio (HR) and Normalized Discounted Cumulative Gain (NDCG). Following the common strategy [24, 36], for each test sequence, all evaluated models predict the scores of 100 candidate products and the top-$K$ products with the highest scores form the final ranked list. HR@$K$ measures whether the ground-truth product is present on the top-$K$ ranked list, while NDCG@$K$ further emphasizes the position of the hit by assigning higher weights to hits at topper ranks. We set $K = \{5, 10\}$ and report the average metrics for all samples in the test set.

*4.1.4 Implementation Details.* We implement the compared methods following the original settings. The embedding dimension $d$ is set to 32 for Amazon datasets and 64 for the JDsearch dataset, and the hidden dimension in feed-forward networks is set to twice the embedding dimension. The number of Siamese Encoder layers $L$ and the number of sessions $N$ are set to $(2, 2)$ for Amazon datasets and $(3, 4)$ for the JDsearch dataset. The effects of $d$, $L$, $N$ are discussed in Appendix A. The weight $\alpha$ assigned to the self-supervised learning loss is set to 0.1, given the results shown in Section 4.3.1. Following [26], we train the model using the Adam optimizer [16] and the warmup-and-decay learning rate schedule. We initialize model parameters using the Xavier initialization [9]. For all models, we employ the default configuration of 100 training epochs and the mini-batch size of 128. Our model is implemented in PyTorch and publicly available[1].

## 4.2 Performance Comparison

We compare UnifiedSSR with search and joint models in the search scenario, and with recommendation and joint models in the recommendation scenario. From the performance comparison shown in Table 2, we have the following observations:

- UnifiedSSR achieves the best performance over all baselines in both search and recommendation scenarios across three datasets. This confirms that the proposed UnifiedSSR effectively addresses the challenges of cross-scenario cross-view user behavior modeling and dynamic user intent discovery, resulting in enhanced capabilities in both two scenarios.

[1](Anonymized) https://anonymous.4open.science/r/UnifiedSSR.

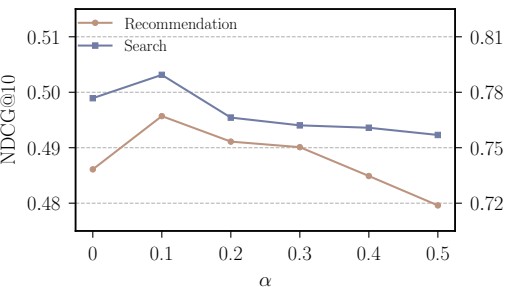

**Figure 3: Performance comparison on Amazon-CL with different settings of self-supervised learning loss weights ($\alpha$).**

- Joint models consistently outperform scenario-specific models on both scenarios, except that SESRec performs slightly worse than FMLP-Rec for recommendation on Amazon-EL. This suggests that joint models have an advantage over scenario-specific models but require the effective incorporation of inherent correlations across scenarios.
- UnifiedSSR-S and UnifiedSSR-R yield competitive performance in their respective scenarios, highlighting the capacity of the designed model architecture for single-scenario user behavior learning. Moreover, UnifiedSSR outperforms UnifiedSSR-S and UnifiedSSR-R in most cases, demonstrating the significance of cross-scenario information sharing during multi-task joint pre-training.

## 4.3 Study of UnifiedSSR

*4.3.1 Impact of Self-Supervised Learning Loss.* As introduced in Section 3.5, the hyper-parameter $\alpha$ in Equation (13) controls the weight of the self-supervised learning loss during training, which guides the intent-oriented session discovery for user intent understanding. To explore the influence of $\alpha$ on the performance of UnifiedSSR, we compare the performance of $\alpha$ over the range of $[0, 0.5]$ at intervals of 0.1. From the results on Amazon-CL shown in Figure 3, we can see that the performance of UnifiedSSR improves as $\alpha$ increases from 0 to 0.1. The performance improvement demonstrates that guided by the self-supervised learning objective, the Intent-oriented Session Modeling module effectively locates and aggregates the intent-oriented semantic sessions, contributing to the dynamic user intent understanding. Besides, the performance becomes worse than $\alpha = 0$ when $\alpha \geq 0.2$ for search and $\alpha \geq 0.4$ for recommendation. This suggests that excessively focusing on intent-oriented session modeling may constrain the capacity for representation learning of user behaviors, leading to a decrease in performance.

*4.3.2 Ablation Study.* To investigate how the various designs impact the performance of UnifiedSSR, we conduct an ablation study considering the following variants: (1) **UnifiedSSR w/o FT**: The model solely undergoes multi-task joint pre-training without any subsequent task-specific fine-tuning. (2) **UnifiedSSR w/o CA**: The multi-head cross-attention sub-layer in the Siamese Encoder layer that encodes the correlation between dual behavior sequences is removed. (3) **UnifiedSSR w/o SE (a)**: The encoder in two branches for the search task share the same architecture but have separate parameters. (4) **UnifiedSSR w/o SE (b)**: The encoder in the product

**Table 2: Performance Comparison with Baseline Methods**

| | JDsearch | | | | Amazon-CL | | | | Amazon-EL | | | |
|---|---|---|---|---|---|---|---|---|---|---|---|---|
| | HR@5 | HR@10 | NDCG@5 | NDCG@10 | HR@5 | HR@10 | NDCG@5 | NDCG@10 | HR@5 | HR@10 | NDCG@5 | NDCG@10 |
| *Search Scenario* | | | | | | | | | | | | |
| HEM | 0.5432 | 0.7590 | 0.2781 | 0.3441 | 0.5504 | 0.6448 | 0.3006 | 0.3298 | 0.5354 | 0.6638 | 0.2864 | 0.3259 |
| ZAM | 0.5547 | 0.7664 | 0.2853 | 0.3501 | 0.5866 | 0.6751 | 0.3238 | 0.3510 | 0.5727 | 0.6931 | 0.3080 | 0.3451 |
| CAMI | 0.3911 | 0.5051 | 0.2929 | 0.3299 | 0.6594 | 0.7539 | 0.5274 | 0.5582 | 0.7118 | 0.7992 | 0.5384 | 0.5669 |
| JSR | 0.8099 | 0.8543 | 0.7347 | 0.7490 | 0.7506 | 0.8060 | 0.6571 | 0.6752 | 0.8197 | 0.8647 | 0.7309 | 0.7455 |
| JSR-Seq | 0.8586 | 0.8781 | 0.8209 | 0.8270 | 0.7565 | 0.7785 | 0.7023 | 0.7088 | 0.8333 | 0.8529 | 0.7980 | 0.8029 |
| SESRec | 0.8809 | 0.9267 | 0.7865 | 0.8019 | 0.7974 | 0.8455 | 0.6977 | 0.7115 | 0.8875 | 0.9125 | 0.8041 | 0.8111 |
| UnifiedSSR-S | 0.9332 | 0.9510 | 0.8856 | 0.8911 | 0.8435 | 0.8784 | **0.7782** | **0.7898** | **0.9091** | **0.9340** | **0.8557** | **0.8628** |
| UnifiedSSR | **0.9551** | **0.9723** | **0.9005** | **0.9057** | **0.8582** | **0.8992** | 0.7757 | 0.7894 | 0.8998 | 0.9304 | 0.8286 | 0.8386 |
| *Improv.* | 8.43% | 4.93% | 9.69% | 9.51% | 7.62% | 6.35% | 11.17% | 10.94% | 1.39% | 1.96% | 3.04% | 3.39% |
| *Recommendation Scenario* | | | | | | | | | | | | |
| GRU4Rec | 0.7514 | 0.8020 | 0.6787 | 0.6949 | 0.4448 | 0.5610 | 0.3194 | 0.3571 | 0.4840 | 0.5840 | 0.3493 | 0.3814 |
| SASRec | 0.7463 | 0.8034 | 0.6585 | 0.6769 | 0.4517 | 0.5526 | 0.3338 | 0.3665 | 0.4973 | 0.6085 | 0.3620 | 0.3983 |
| FMLP-Rec | 0.7578 | 0.8054 | 0.6935 | 0.7089 | 0.4556 | 0.5802 | 0.3229 | 0.3634 | 0.5268 | 0.6473 | 0.3846 | 0.4236 |
| JSR | 0.7699 | 0.8174 | 0.7013 | 0.7162 | 0.4853 | 0.6021 | 0.3636 | 0.4011 | 0.5344 | 0.6561 | 0.3880 | 0.4274 |
| JSR-Seq | 0.7876 | 0.8340 | 0.7156 | 0.7304 | 0.5579 | 0.6655 | 0.4307 | 0.4655 | 0.5577 | 0.6725 | 0.4167 | 0.4543 |
| SESRec | 0.7878 | 0.8361 | 0.7169 | 0.7322 | 0.5013 | 0.5932 | 0.3958 | 0.4252 | 0.5186 | 0.6337 | 0.3831 | 0.4210 |
| UnifiedSSR-R | 0.7828 | 0.8343 | 0.7108 | 0.7272 | 0.4628 | 0.5672 | 0.3471 | 0.3807 | 0.5149 | 0.6439 | 0.3680 | 0.4088 |
| UnifiedSSR | **0.8482** | **0.8983** | **0.7586** | **0.7749** | **0.5941** | **0.7004** | **0.4608** | **0.4957** | **0.6036** | **0.7184** | **0.4564** | **0.4933** |
| *Improv.* | 7.67% | 7.44% | 5.81% | 5.82% | 6.48% | 5.24% | 6.99% | 6.49% | 8.25% | 6.82% | 9.54% | 8.59% |

\* The best results are in **bold**, the second best results are underlined.
\* *Improv.* stands for the performance improvement of UnifiedSSR over the best-performing baseline methods.

**Table 3: Performance Comparison on Amazon-CL with UnifiedSSR Variants**

| | Search | | Recommendation | |
|---|---|---|---|---|
| | HR@10 | NDCG@10 | HR@10 | NDCG@10 |
| UnifiedSSR | 0.8992 | 0.7894 | 0.7004 | 0.4957 |
| w/o FT | 0.8825 | 0.7654 | 0.6697 | 0.4640 |
| w/o CA | 0.8920 | 0.7692 | 0.7010 | 0.4913 |
| w/o SE (a) | 0.8762 | 0.7701 | 0.6900 | 0.4874 |
| w/o SE (b) | 0.8896 | 0.7782 | 0.6916 | 0.4866 |
| w/o ISM (a) | 0.8941 | 0.7800 | 0.6956 | 0.4910 |
| w/o ISM (b) | 0.8901 | 0.7697 | 0.6913 | 0.4854 |

branch in two tasks share the same architecture but have separate parameters. (5) **UnifiedSSR w/o ISM (a)**: Instead of learning to extract intent-oriented sessions, the sequences are split into $N$ sessions based on largest $(N-1)$ time intervals. (6) **UnifiedSSR w/o ISM (b)**: The Intent-oriented Session Modeling module for intent-oriented session enhancement is removed.

Table 3 illustrates the experimental results comparing UnifiedSSR and its variants in terms of HR@10 and NDCG@10 on Amazon-CL. From Table 3, we have the following observations:

- UnifiedSSR w/o FT exhibits a reasonable performance drop in both scenarios compared to UnifiedSSR, yet it can still achieve competitive performance with baselines solely through pre-training. This validates the robust representation capability of UnifiedSSR based on multi-task joint learning.

- UnifiedSSR w/o CA, w/o SE (a), w/o SE (a) reduce the extent of information sharing from different perspectives. The performance decreases in these variants indicate the importance of cross-scenario cross-view information sharing for joint learning of user behaviors in both search and recommendation.

- UnifiedSSR w/o ISM (a) performs better than UnifiedSSR w/o ISM (b) in both scenarios. The difference between these two variants is that the former enhances behavior sequence modeling with time-interval based sessions while the latter does not use any session information. UnifiedSSR further outperforms UnifiedSSR w/o ISM (a), verifying that UnifiedSSR effectively leverages dynamic user intent through intent-oriented session modeling, thereby enhancing the model performance in both scenarios.

## 5 CONCLUSIONS

In this work, we proposed a unified framework for joint learning of user behaviors in both search and recommendation scenarios. Specifically, UnifiedSSR adopted the dual-branch architecture that encodes the pair of product history and query history in parallel in the search scenario, and deactivates the query branch to adapt to the recommendation scenario. UnifiedSSR effectively shared information cross-scenario (*i.e.*, search and recommendation scenarios) and cross-view (*i.e.*, interacted products and issued queries in the search scenario), while simultaneously modeling the dynamic user intent through the intent-oriented session discovery guided by two self-supervised learning signals. Extensive experiments on three public datasets demonstrated the effectiveness of UnifiedSSR.

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

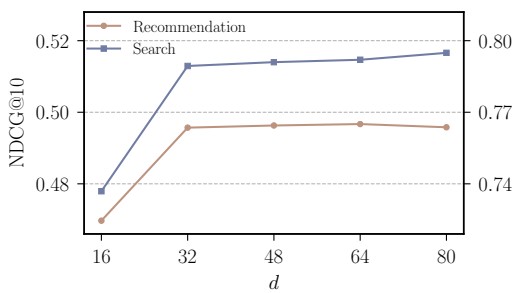

Figure 4: Performance comparison on Amazon-CL *w.r.t.* NDCG@10 with different settings of embedding dimensions ($d$).

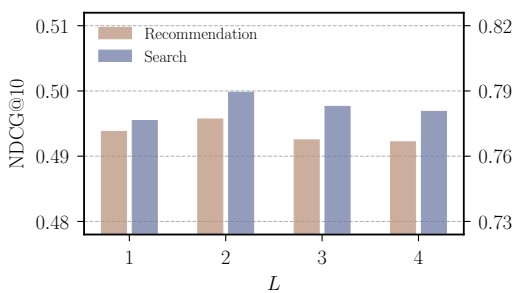

Figure 5: Performance comparison on Amazon-CL *w.r.t.* NDCG@10 with different settings of Siamese Encoder layer numbers ($L$).

## A PARAMETER ANALYSIS

### A.1 Impact of Embedding Dimension

We conduct experiments to analyze the impact of the embedding dimension (*i.e.*, $d$) in UnifiedSSR. As an example, in the Amazon-CL dataset, we vary $d$ from 16 to 80 in increments of 16. Figure 4 illustrates the experimental results *w.r.t.* NDCG@10 on two tasks. Based on Figure 4, we can observe a significant drop in performance when $d = 16$ for both tasks, indicating that it is insufficient to encode the contextual information. As the embedding dimension increases, the performance first exhibits substantial improvement, followed by a gradual stabilization and occasional slight declines. Considering the trade-off between cost and performance, we set the default $d = 32$ for Amazon datasets and $d = 64$ for the JDsearch dataset.

### A.2 Impact of Siamese Encoder Layer Number

The Siamese Encoder encodes the correlations both within each behavior sequence and across dual behavior sequences. The encoded representations at all positions in both sequences are essentially projected into a common space, where similar behavior patterns are close to each other. Here we analyze how the number of Siamese Encoder layers (*i.e.*, $L$) impacts the model performance in two scenarios. To achieve this, we conduct experiments with varying settings of $L$ ranging from 1 to 4. Figure 5 illustrates the performance comparison on Amazon-CL. We observe that the performance consistently peaks at $L = 2$ on both search and recommendation tasks, followed by a gradual decline as $L$ increases. This decline may be attributed

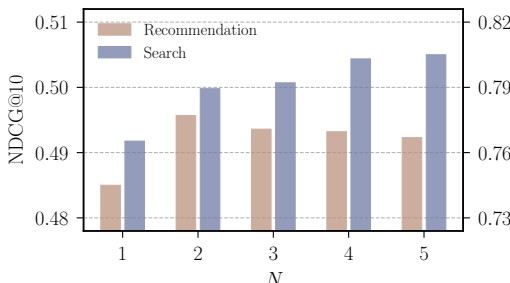

Figure 6: Performance comparison on Amazon-CL *w.r.t.* NDCG@10 with different settings of session numbers ($N$).

to the overfitting problem. Based on the experimental results, we set $L = 2$ as the default for Amazon datasets and $L = 3$ for the JDsearch dataset, where the model performs best.

### A.3 Impact of Session Number

The number of sessions $N$ plays a crucial role in UnifiedSSR. When $N$ is set too large, it becomes challenging to locate semantic sessions with shorter initial lengths. Conversely, if $N$ is set too small, sessions with longer initial lengths are more likely to include interactions with low correlation, thereby introducing unwanted noises. Therefore, here we investigate how the number of sessions $N$ affects the performance of UnifiedSSR. In particular, we vary $N$ within the range $[1, 5]$ and present the results in Figure 6. We can observe that the model performance steadily improves in the search task as $N$ increases, while the performance reaches its peak at $N = 2$ in the recommendation task. One possible reason for the different performance trends between the two scenarios is that, without an explicit query, user intent in the recommendation scenario tends to be ambiguous, resulting in less distinguishable intent-oriented sessions, and thus higher values of $N$ may unnecessarily capture semantically meaningless sessions, undermining the performance. Based on the experimental results, we set default $N = 2$ for Amazon datasets and $N = 4$ for the JDsearch dataset.

