# OpenReview forum: "UnifiedSSR: A Unified Framework of Sequential Search and Recommendation"
_ACM.org/TheWebConf/2024/Conference — TheWebConf24 Oral_

### Official Review · Reviewer_ABau · 2023-11-23

**Novelty:** 5
**Technical Quality:** 5

**Review:**

- The paper proposes a framework that considers user interactions with products in recommendation scenarios and user-issued queries in search scenarios as distinct types of user behaviors. This framework aims to encode the history of interacted products and issued queries in parallel, allowing for cross-scenario modeling.
- UnifiedSSR employs a dual-branch architecture with a product branch and a query branch. These branches share parameters to transform the sequences into a common latent space, enabling the model to learn user behavior patterns across both views. The architecture allows for the deactivation of the query branch in the recommendation scenario, facilitating cross-scenario joint learning.
- The insight of this paper is highly effective. Search and recommendation are the two scenarios with the most user interactions on websites, where user behaviors in both scenarios often share similar objectives. Unifying the learning of these two behaviors should aid in enhancing both search and recommendation outcomes, representing a promising research direction.

**Questions:**

- this scenario, user queries in search are expected to be significantly fewer than interactions in the recommendation system. How does this model align the data between search and recommendation given such a data bias?

**Reviewer Confidence:**

4: The reviewer is certain that the evaluation is correct and very familiar with the relevant literature

**Scope:**

4: The work is relevant to the Web and to the track, and is of broad interest to the community

---

### Official Review · Reviewer_47Lo · 2023-11-23

**Novelty:** 5
**Technical Quality:** 6

**Review:**

The authors introduce a unified framework based on three-way shared encoder parameters, i.e., query in search, products in search, and products in recommendations. Additionally, the authors incorporate an intent-oriented session module. The authors show the merits of their approach on two datasets (one with two sub datasets) in the search and recommendation scenarios, where their solutions consistently perform best. While the premise of the paper is relevant, the presented work is not entirely coherent with two parts being mostly independent (i.e., do not support each other). As a result, neither parameter-sharing nor session-intent modeling is fully explored and thus lacks depth. I suppose two independent publications, each with additional details and experiments, would better suit the research.

Strengths:
S1 - Three-way parameter sharing is interesting.
S2 - The paper is well written.
S3 - Broad evaluation with promising results.

Weaknesses:
W1 - Joint learning of search and recommendation behavior not fully explored.
W2 - Session-intent modeling only weakly supports UnifiedSSR.
W3 - Several details are not clearly communicated.

**Questions:**

Regarding a "Unified framework of Sequential Search and Recommendation":
Q1a: Why do you exclude side-information? I do not see simplicity as a good argument, but a unified framework should likely account for that.
Q1b: Why do you model search and recommendation separately? For instance, a user might first follow the recommendations for inspiration and afterward switch to search for more directed navigation.

Regarding Figures:
Q2a: Figure 1 could be reordered for clarification. The caption states that the vocabulary is also shared, which does not make sense to me.
Q2b: Figure 2 is not consistent, as each subfigure shows input and output differently.
Q2c: How is Figure 2b different from a Transformer decoder layer? It seems to be missing an "Add & Norm" between the self and cross-attention. Why is that?
Q2d: Figure 2c is not clear to me. For instance, what do the arrows between the feed-forward and projection mean?

Regarding Datasets:
Q3a: What is #Samples-S and -R exactly? Number of sessions considered for search and recommendations?
Q3b: What is the average number of (distinct) products and sessions per user? What is the average number of (distinct) products per session? Can you further clarify #QueryWords?

Results:
Q4a: Lines 707-711 should be further clarified regarding how the data differs. Apparently, doing so leads to a drop in performance.
Q4b: Section 4.3.1 suggests that alpha only drops at 0.4 for the recommendation scenario, but Figure 3 suggests a drop after 0.1.
Q4c: Why is there no UnifiedSSR w/o SE (c), where the sharing is completely removed? Similar to UnifiedSSR w/o ISM (b)
Q4d: Why is the ablation study performed on Amazon-CL, which has the worst performance of the three?
Q5d: Even with parts disabled, the results are still higher than the baseline. Have you considered excluding multiple subparts, e.g., by incremental exclusion? It would be interesting to see when it becomes comparable to the baseline models. Also, naming could be improved by underlying the respective letters.
Q6d: The benefit of the ISM does not seem very high. Have you looked into that?

**Ethics Review Description:**

-

**Reviewer Confidence:**

4: The reviewer is certain that the evaluation is correct and very familiar with the relevant literature

**Scope:**

4: The work is relevant to the Web and to the track, and is of broad interest to the community

---

### Official Review · Reviewer_sLJh · 2023-11-27

**Novelty:** 5
**Technical Quality:** 4

**Review:**

**Summary:**
This paper proposed a unified framework for joint learning of user behaviors in both search and recommendation scenarios. The model was designed to incorporate cross-view and cross-scenario association of user behaviors and provide a better understanding of user behavior patterns. They also designed an intent-oriented session modeling to capture users’ intents and enhance user behavior modeling.

**Strength:**

1. The paper is well-presented and easy to follow, and the investigated topic of joint modeling for both search and recommendation is interesting and will be of interest to researchers in both search and recommender systems.
2. The motivation of this research is clearly described, and the challenges addressed in this paper were introduced in a clear manner with adequate review.
3. The presentation of the proposed UnifiedSSR and each component is clear.
4. To validate the effectiveness of the proposed models, experiments were conducted on three public datasets in different domains, showing that the proposed framework can achieve better performance than baseline methods in both search and recommendation scenarios.

**Suggestions for improvements:**

For the dataset processing, the paper stated “For both search and recommendation data, the sequences of each user are chronologically ordered and divided into subsets for multi-task joint learning and task-specific learning in an 8:2 ratio. The multi-task joint learning set is used for model pre-training, while the task-specific learning set is further split into training, validation, and test sets.” It is not very clear how the authors divide the dataset for both search and recommendation scenarios. It is better to explicitly describe in the paper (section 4).

For the evaluation, I wonder if the evaluation was conducted on the dataset one time or multiple times. For the comparison with baseline methods, it is better to add statistical analysis to see the significance of improvement.

**Questions:**

As stated above, I have questions about the evaluation:
How many tests were conducted on one dataset and how the datasets were divided into train, validation, and testing for both scenarios and why?

**Reviewer Confidence:**

2: The reviewer is willing to defend the evaluation, but it is likely that the reviewer did not understand parts of the paper

**Scope:**

3: The work is somewhat relevant to the Web and to the track, and is of narrow interest to a sub-community

---

### Official Review · Reviewer_JCLG · 2023-11-27

**Novelty:** 4
**Technical Quality:** 5

**Review:**

In this paper, the authors propose a Unified Framework of Sequential Search and Recommendation (UnifiedSSR) that facilitates joint learning of user behavior history in both search and recommendation scenarios. It incorporate cross-view and cross-scenario associations of user behaviors, providing a comprehensive understanding of user behavior patterns. And the Intent-oriented Session Modeling module can infer intent-oriented semantic sessions from the contextual information in behavior sequences.
The main contributions are:
1. This paper introduces a novel UnifiedSSR framework, utilizing a dual-branch architecture with shared parameters. This framework enables the joint learning of cross-scenario and cross-view user behaviors. The dual-branch network is designed to encode interacted product history and issued query history in parallel, allowing for cross-scenario modeling by deactivating the query branch for the recommendation scenario.
2. This paper proposes an Intent-oriented Session Modeling module, aiming to enhance user behavior modeling by capturing dynamic user intent. This module leverages self-supervised learning signals to encourage intent-oriented session discrimination within each behavior sequence and intent-oriented session alignment between dual behavior sequences.

Pros:
-	The equations presented effectively illustrate the relationships between different components, enabling a clear understanding of how the proposed framework integrates diverse user behaviors.
-	The authors cleverly use a parameter shared Siamese Encoder that comprehensively captures the correlations both within and between dual behavior sequences.
-	The jointly loss is technically sound and are shown useful for the performance.

**Questions:**

-	Yuqi Gong, et al. also propose an unified Framework of Search and Recommendation. This paper should also be analyzed and compared

Yuqi Gong, et al. 2023. An Unified Search and Recommendation Foundation Model for Cold-Start Scenario. In Proceedings of the 32nd ACM International Conference on Information and Knowledge Management (CIKM '23)

**Reviewer Confidence:**

3: The reviewer is confident but not certain that the evaluation is correct

**Scope:**

4: The work is relevant to the Web and to the track, and is of broad interest to the community

---

### Decision · Program_Chairs · 2024-01-22

**Decision:**

Accept (Oral)

**Comment:**

All reviewers found merits in the submission and are happy about the authors' rebuttal information. I still recommend the authors to follow their suggestions to improve their work if it could be finally accepted.